# The Socio-Linguistic Adaptation of Migrants: The Case of Oralman Students' Studying in Kazakhstan

**Fatima Valieva [1],*, Jannat Sagimbayeva [2], Dina Kurmanayeva [2] and Gulzhakhan Tazhitova [2]**

[1]    Peter the Great St. Petersburg Polytechnic University, St-Petersburg 195251, Russia
[2]    L. N. Gumilyov Eurasian National University (ENU), Nur-Sultan 010008, Kazakhstan
*    Correspondence: jf.fairways@mail.ru; Tel.: +7-9219759799

**Abstract:** This article examines different aspects of the oralman students' socio-linguistic adaptation. Scientific research has identified various obstacles oralmen face when studying at Universities in Kazakhstan, especially in the context of the English language studying process. The data used in this paper explores certain peculiarities of oralman students' adaptation to new educational environments, their attitude towards English, and their difficulties in advancing in foreign language learning. The electronic questionnaires, which included the Likert scale, multiple choice and open-ended questions, were offered to 600 students, while interviews were conducted with 90 students in the last stage of the research. Descriptive statistics and one-way analysis of variance were used to analyze the data. The scientific findings indicated some unexpected difficulties in the oralman students' socio-linguistic adaptation, which decreased their motivation for learning the foreign language and reduced effectiveness of the teaching. However, certain noticeable discrepancies between the anticipated and real educational context were revealed. The interview and data analysis made it possible to uncover cultural, linguistic, social, and psychological problems. The scientific research suggests that the incorporation of some integrative effective methods and techniques into English classes would be useful for improvement of the oralman students' English language skills.

**Keywords:** oralman students; English; educational environment; adaptation problems; difficulties in studying; socio-linguistic adaptation; migrants' adaptation

---

## 1. Introduction

Several studies have aimed to explore various challenges of modern higher education, and most of these are concerned with specific subjects such as research on international education [1], international students [2], and online learning [3]. Higher education investigators have presented the following topics in their research: mobility of student and staff; internal and external influences of higher education systems on each other; internationalization of teaching, learning and research; new teaching strategies and technologies; knowledge transfer; modes of cooperation and competition; national and sub-national policies regarding the international dimension of higher education, and others [4,5].

International academic mobility has become an integral part of modern higher education. On the one hand, it is beneficial for host universities to open frontiers and enroll international students in their study programs, as it helps to obtain more financial investment for further development. On the other hand, universities have had to develop new strategies and technologies to provide a successful, effective sociocultural and psychological adaptation. Students who live and study abroad for long periods are not only introduced to unfamiliar cultures, but sometimes also find themselves immersed in them. Unfortunately, not everyone adapts easily to the unfamiliar realities that surround them.

The language barrier is considered by most researchers of cross-cultural adaptation to be a universal obstacle to integration into host culture environments [6]. However, in some cases even a high level of language proficiency does not guarantee productive and successful communication. A study of foreign postgraduate students at a UK university, for example, revealed that students who had been successfully using English in their academic environment still faced difficulties with speaking English in real-world settings in the UK [7]. Such a phenomenon is generally connected to an inability to communicate freely with native and non-native foreign language speakers. It results in negative emotional responses such as apprehension, anxiety, shame, fear of being tested and evaluated, and low self-esteem [4,6].

Content analysis of the latest scientific research studies has revealed that psychological, socio-cultural, and linguistic adaptation is understudied for expatriates and first-generation migrants; culture-learning factors (cultural distance and interaction) are overlooked for first-generation migrants, as are family factors for international students and stressors for expatriates [6].

Oralman students' problems in higher education in Kazakhstan became one of the most important and significant issues, intertwined with the difficult historical period of the country's development. Oralmen or oralman students are students whose grandparents or parents left Kazakhstan during the 1920–1930 period due to violent collectivization, hunger, and repressions, which brought suffering to most of the Kazakh population. The process of resettling ethnic Kazakhs to the historical homeland gained momentum after Kazakhstan gained independence and in 1991 significant changes began in all spheres of public life and in the growth of Kazakhstan's international prestige. Since then, Kazakh oralmen have been returning to their native land. Historian A. Suvorov divided the actual flow of oralman people into two types: regulated, when the relocation of the people is carried out at the expense of the republican budget, and spontaneous, when oralmen independently move to Kazakhstan. For years after Kazakhstan gained independence, the number of oralman people grew significantly and currently totals approximately one million. Consequently, the ethnic Kazakhs are not only essential as an influential force in Kazakhstan, but are also an important factor in the social development of certain regions, primarily with regard to defining the political, economic, and cultural spheres of life [8,9].

Because of the constantly increasing flow of oralmen to Kazakhstan, the number of oralman students in higher institutions of higher education has also increased. This is due to the government's special attention to oralman students. A quota of scholarships is assigned to them as they enter the universities of Kazakhstan. Thus, approximately 1500 young Kazakhs from Russia, Uzbekistan, Mongolia, China, Tajikistan, Turkmenistan, Kyrgyzstan, Turkey, and Iran arrive in Kazakhstan annually to obtain higher education [10]. The largest group comprises young people who moved to Kazakhstan from Uzbekistan, China, and Mongolia. For example, in the last five years, 2363 young oralmen arrived at the four leading universities of Astana (Table 1).

**Table 1.** Enrolment of immigrant students at the universities of Astana.

| Universities of Astana | Total | Uzbekistan | China | Mongolia | Other Countries |
|---|---|---|---|---|---|
| Astana Medical University | 916 | 375 | 275 | 203 | 121 |
| Kazakh National University of the Arts | 754 | 203 | 124 | 77 | 49 |
| L.N. Gumilyov Eurasian National University | 537 | 253 | 197 | 142 | 54 |
| S. Seifullin Kazakh Agro Technical University | 499 | 132 | 75 | 58 | 25 |
| TOTAL | 2363 | 963 | 671 | 480 | 249 |

(Taken from http://rus.azattyq.org/content/kazakhstan-repatriants-statistics/).

English is the lingua franca for educational subjects at the Republic level and English is given the same role in the general subject register as the Kazakh and Russian languages. Since oralman students have lived in different societies with different cultures, they have had to adapt to a very new life context in Kazakhstan. Consequently, some specific difficulties of social, psychological, and linguistic origin need to be considered. Oralman students do not learn or master English in identical ways because

the process is influenced by various positive or negative factors, from individual differences among students to grammatical differences. This process depends on external and internal factors, as well as the individual attitudes of students, their fears, or established distinctions based on social or economic status. According to the educational standard of Kazakhstani Universities, students must be at an intermediate level of English. The main problem for the teaching process is that the oralman students only have a basic level of English and additional courses are not organized for them. The main point here is that there are some psychological and methodical peculiarities in teaching English to oralman students that distinguish them from local students [11].

The research is relevant due to requirements of modern Kazakhstani society and Kazakhstan's current educational priorities. The aim of this research was to identify distinct features of the oralman students' adaptation in the educational environment of Kazakhstan. This entailed the determination of the level of linguistic competence of the oralman students, the problems Kazakhs face in learning English, the obstacles that teachers face when teaching this category of students, and ways to improve these students' knowledge of English.

In the study, researchers conducted a questionnaire presented to 600 oralman students, which comprised 3 sections: level of proficiency in languages in the form of a test, social and psychological characteristics, and difficulties in studying English at an institution of higher education. By holding interviews with 90 oralman students and 10 English teachers, the authors identified the difficulties they face in the studying and teaching process. These difficulties made it possible to identify the need for creating new ways of learning and teaching of English to oralman students.

### 1.1. Literature Review

Attempts to explain what predetermines success and failure of international experience in its multicultural diversity have resulted in a vast body of publications on students' socio-cultural and socio-linguistic adaptation. The articles discuss general problems of socio-cultural, socio-linguistic, and psychological adjustment [12–15]. Such variables as language proficiency, host national contact, cultural distance, perceived discrimination, and others influence the level of students' social adjustment to the host culture and their psychological well-being [16,17].

Of particular interest are studies examining the problems of the adaptation of migrants (immigrants, foreigners) [18–22]. Researchers address contradictions between students' level of communicative competence, perceptions of educational information and the requirements of higher education. Studies also examine the academic stereotypes of the country of training [18]. Personal problems involve dissatisfaction with the distribution of social roles, the impossibility of sufficiently manifesting oneself as an individual, the impossibility of satisfying requirements and fulfilling a significant purpose. Isolation, discomfort regarding the awareness of distinctions between cultures, confusion about value orientations, a high level of uneasiness, insufficient control of behavior, aggression and conflict are often identified among foreign students and immigrants [20,21]. Language adaptation plays an important role in the process of socialization, which directly affects the success of communication. Among these difficulties, scientists emphasize the significance of insufficient knowledge of the language in a new socio-cultural environment, the inability to define various social roles that are assumed by participants, and ignorance of implications that are clear to native speakers [19]. According to scientists, the acquisition of socio-linguistic competence in another language is a long and difficult process and includes the understanding of social values, being the cornerstone of language use in a society [22,23]. This study presents a set of useful techniques that will help students express themselves properly in various social contexts in line with the development of social and linguistic competences.

As far as subject literature is concerned, works devoted to the language training of migrants in a multilingual environment are of particular relevance. Researchers consider the integration of heterogeneous groups to be dependent on language cooperation, accustoming students to the multilingual environment, encouraging the development of multilingual skills, and improving tolerance [24]. Egbert and Simich-Dudgeon (2001) noted foreigners' difficulty in assimilating English

grammar. Complex tense forms and the use of abstract words complicate verbal interactions necessary for the understanding of text in written communication [25]. Schmid (2014) considered the influence of accent and pronunciation on the clarity and reliability of listening comprehension [26]. Chrabaszcz (2014) studied the influence of the grammatical structure of the native language on assimilation and the use of the grammatical phenomena of English in oral and written language [27]. Rickheit and Strohner (2010) considered communicative competence to be the ability of a language user to organize his speech behavior adequately in communication situations given the communicative purpose, intention, social status, roles of communicants, and the communication situation according to the socio-linguistic norms and attitudes of the national linguistic and cultural community [28].

Similar to the studies mentioned above, the present research follows, in investigating socio-linguistic adaptation of migrants, that one of the most influential models applied in various studies is the bi-dimensional adaptation model proposed by Ward and colleagues [29], which views adaptation as a bi-dimensional phenomenon.

The first dimension, based on socio-cultural adaptation, refers to the behavioral domain. The second dimension connected with psychological adaptation, "refers to one's well-being within the new culture and is underpinned by the process of coping with the stress of intercultural transition" [6] (p. 2).

The immigration of ethnic Kazakhs to their historical homeland assumes that these oralmen can overcome the difficulties of the adaptation and integration processes. These difficulties include social problems (oralman students have feelings of fear because they are in an uncertain situation and consider themselves strangers) and problems with adapting to the new cultural environment (uncertainty about and failure to understand what is accepted) [30]. Behavioral models were created by addressing a difficult set of problems whose solutions require careful study. Overcoming a language barrier presents many problems (e.g., fear, hesitation, shame about poor pronunciation) [31]. Successful oralmen integration into Kazakhstani society depends on factors that include identifying the language situation that has developed in the territory of Kazakhstan, the language policy of the Republic of Kazakhstan, and the activity of the government bodies of Kazakhstan in relation to ethnic Kazakh immigrants. Other factors include establishing the role of the mass media and identifying the role of education. Thus, methods of teaching English in higher education that consider various components are being developed. However, methodical support of this educational work in higher educational institutions for oralman students, in the form of special programs and manuals, is required.

### 1.2. English in Kazakhstan

Demand for the knowledge of languages in Kazakhstan has increased due to public, political, and economic changes in the last twenty years. These changes have had a noticeable impact on the development of various fields of life including business, transport, trade, diplomacy, education, and medicine. Successful and dynamic integration of Kazakhstan into the world community is possible by acquiring new knowledge through the mastery of dominant languages. It is the implementation of polylingualism in education, which will strengthen the competitiveness of Kazakhstan. The main purpose of teaching the three languages is to provide opportunities to learn, without any enforcement, and it is the basis for resolving the language issue in Kazakhstan. Thus, the pursued language policy guarantees the observance of rights of all ethnic groups and provides a free choice of language for communication, education, and implementation of creative needs.

Language policy in Kazakhstan is based on the idea of the cultural project, the "Linguistic trinity" initiated by the head of state. The essence of the project is to study Kazakh as the state language, Russian as the language of international communication, and English as the language of successful integration into the global economy. Phased implementation of the project began in 2007. The main strategic steps and tasks in the framework of the "Linguistic trinity" cultural project are studying English and other foreign languages to create a multilingual educational environment and develop the intellectual potential of the country by initiating its citizens into the values of language and culture [32].

In light of the project's implementation, English as a foreign language has a leading position in the country's educational institutions. Children start learning English in kindergarten. Some school subjects are taught in English, thereby not only forming and developing language skills, but also broadening students' substantive knowledge. Multilingual groups are formed at universities.

English is the language of multicultural communication, which is needed for improving all of the above-mentioned spheres (and others). Moreover, competence in English can be the defining element of competitiveness of young specialists in the labor market, both in the country and abroad. Furthermore, English is designated as a condition of successful entry into the global economy and has begun to be considered one of the main priorities of state policy. Program documents regarding the need for a universal solution of issues relating to teaching English and other languages as a means of international communication have been developed in recent years.

Knowledge of English has many benefits in Kazakhstan. First, one can access new information in English that is important for their studies or work. Secondly, it is connected with the opportunity to find more prestigious jobs or the possibility of career development. Besides that, knowledge of English facilitates integration into the world community, opening doors to international companies, allowing communication with people from other countries, facilitating sharing of achievements and experience, and promoting cooperation with foreign clients and companies.

English is taught at all levels of education in Kazakhstan. It is taught first in primary school, where the content of the program is aimed at developing learners' interest and basic language skills. In middle school, the teaching of English corresponds to Levels A1 and A2 by the Common European Framework Reference for Languages (CEFR). Students are supposed to complete language learning at the B1 level in high school. Students of non-linguistic specialties finish their first year at the B2 level.

The modern concept of teaching a foreign language in non-linguistic departments provides students, who are eager to engage in intercultural communication, with a chance for personal development and self-realization, while also satisfying the labor market's need for specialists able to continue their education and professional activities in a foreign language environment [33–36].

The monitoring of language teaching quality is based on state educational standards and on the concept of foreign language education in the Republic of Kazakhstan that imply European standard-based multi-level education with a gradual expansion of the scope of foreign language application at the university level. The latter criterion is especially important. Groups with a focus on learning English regardless of specialty were created in 2012 in 20 non-linguistic institutions of higher education in the Republic of Kazakhstan.

The concept of the development of Kazakhstani foreign language education is based on the program and standards of the "Common European Framework of Reference for Languages" [37], which designates 6 levels of proficiency: A1, A2, B1, B2, C1, C2. High school students must beat least at a level of B1 or B2 of English. For this purpose, the educational process is to be updated in terms of its program, methodology, and technology. For a number of years, the main foreign language textbooks have been selected from the offer of foreign publishing houses such as Cambridge Publishing, Oxford, Macmillan, and others. These books enable teachers to teach students by using authentic materials that equip them with knowledge of English-speaking countries as well. The educational process excludes the adapted and outdated textbooks from local publishing houses. It is obvious that creative selection of material, as well as use of all possible means of learning, such as audio, video, and web resources, is considered, in order to motivate students' interest in independent learning and communication. The program also includes LAP (language for academic purposes) and the special professional LSP (language for special purposes) programs. Compulsory state standards for higher education require foreign language to be taught in the first and second year of study at the university level.

The standard training program/curriculum, "A Foreign Language for Non-Linguistic Specialties of Higher Education Institutions for a Bachelor's Degree" (2010) [38], is the result of adapting the international model of foreign language education to the conditions of the Republic of Kazakhstan. English in non-linguistic specialties is studied as a general subject over three semesters. During the first

two semesters, students are engaged for three hours per week. In the third semester, students study professionally oriented English for two hours per week. The main objective of teaching English according to the State Compulsory Educational Standard of the Republic of Kazakhstan is to increase students' communicative, sociolinguistic, and pragmatic competences [38].

Communicative competence indicates that the student possesses the ability to understand authentic speech at a pace natural to native speakers and can practically apply the acquired theoretical knowledge in the field of phonetics to the communication process considering the specifics of the pronunciation of the language (the correct phonetic registration of prepared and unprepared statements of different degrees of complexity) [39]. The development of sociolinguistic competence includes learning about ethical and moral standards of behavior, non-verbal means of communication (mimicry, gestures), the ability to recognize markers of the speech characteristics of humans (social status, ethnicity, and others) at all levels of language, and the ability to recognize the linguistic markers of social relations and use those markers appropriately (e.g., the formulas of greeting, farewell, and emotional exclamation).

The development of pragmatic competence involves the ability to communicate in various communicative situations and to use language to achieve one's communicative purposes and the desired effects (e.g., expression of opinion, agreement/disagreement, desire, and request). During the course of study, socio-personal competences are also developed that involve mastering the skills of cross-cultural communication, maintaining adequate social and professional contacts, being guided by the principles of tolerance and ethno-cultural ethics, and developing respect for foreign-language cultures.

## 2. Materials and Methods

### 2.1. Participants

This research was conducted at the departments of foreign languages of four universities: Eurasian National University, Kazakh National University of Arts, Astana Medical University, and the Kazakh Agro Technical University. These universities differ from each other, but English is taught in all of them during the first year of study with the same number of hours devoted to it. The decision to focus on these universities was dictated by the fact that they are located in Astana and about 70% of oralman students were studying there. As illustrated in Table 2, 600 oralman students participated in the study. Among them, 195 respondents were studying at the Eurasian National University, 145 at the Kazakh National University of the Arts, 140 at the Astana Medical University, and 120 format the Kazakh Agro Technical University. Of these participants, 64.2% were men and 35.8% were women. The age of the oralman students varied from 17 to 23 years of age (Table 2).

**Table 2.** Participants' general profile.

| Country of Origin | Number of Participants | % | Age | Gender (M/F) |
| --- | --- | --- | --- | --- |
| Uzbekistan | 235 | 39.12 | 17–21 | 137/98 |
| Mongolia | 171 | 28.5 | 17–23 | 116/55 |
| China | 170 | 28.3 | 17–23 | 111/59 |
| Other countries | 24 | 4.08 | 17–21 | 21/3 |
| Total | 600 | 100 | | |

Moreover, ninety oralman students and ten EFL teachers agreed to participate in the interview: 29 oralman students from the Eurasian National University, 21 students from the Kazakh National University of the Arts, 20 students from the Medical University, and 20 students from Kazakh Agro Technical University. They were interviewed individually and each interview lasted 25 to 30 min. The interview questions related to identifying oralman students' difficulties in studying English. As for teachers, all of them have had experience in teaching oralman students. Four teachers were from the Eurasian National University, six from the Astana Medical University, the Kazakh National University

of the Arts, and the Kazakh Agro Technical University. One teacher was pursuing a master's degree in teaching English as a foreign language; however, she has a lot of experience teaching oralman students. Five teachers were well qualified and experienced, and four teachers were at the start of their teaching career. This interview included eight questions and was conducted in Kazakh and Russian, lasting 15 min for each respondent.

The following research questions guided this study:

1.  What is the English level of oralman students' studying at universities of Astana?
2.  What difficulties do oralman students and English teachers have in studying and teaching English?
3.  What is the oralman students' preference towards languages and the language situation in Kazakhstan?
4.  What methods should be incorporated to raise these students' level of English?

*2.2. Data Collection and Analysis*

Data collection methods included a questionnaire and interviews with teachers and oralman students. The questionnaire was completed by oralman students after their classes in the department offices. Oralman students had to answer three sections of questions, and it took 40 min to answer them.

The purpose of the questionnaire was to develop an overall picture of the role of English in the respondents' lives, and to define the social, psychological, and linguistic difficulties of studying English. The questionnaire comprised 80 questions, subdivided into 3 sections (Table 3):

Section 1. Level of proficiency in languages (in the form of a test).
Section 2. Social and psychological characteristics.
Section 3. Difficulties with studying English in institutions of higher education.

**Table 3.** Breakdown of questionnaire items.

| Eliciting: | Question N |
| --- | --- |
| Participants' background | 1–7 |
| Level of proficiency in languages (in the form of a test) | 8–52 |
| Social and psychological characteristics | 53–67 |
| Difficulties with studying English in institutions of higher education. | 68–80 |

Participants' background included gender, age, education, nationality, the duration and the place of residence in Kazakhstan, and biographical data. The first section determined the level of oralman students' language proficiency and compliance with language requirements at the university. Testing included lexical and grammatical tasks, speaking, reading, writing, and listening. Lexical and grammatical questions were presented with five answers, only one of which was correct. Tasks covered grammatical topics that are included in the model curriculum for Foreign Language. Furthermore, respondents were asked to rate their own oral English language proficiency and comprehension, fluency, vocabulary, pronunciation, and grammar. Six hundred students participated in the placement test.

The second section comprised social and psychological characteristics. This section revealed preferences of the participants concerning the use of languages and the participants' opinions regarding the use of English in Kazakhstan. It also included questions regarding the language situation in Kazakhstan and opinions of the respondents regarding the development of languages in Kazakhstan and spheres of language application. The peculiarities of students' adaptation were estimated with the help of several statements concerning emotional intelligence, resilience and social support. The second section captured the specificities of individual sustainability through the authorized Resilience Scale. It embraces some statements devoted to social support availability and necessity, and EIS based on emotional competencies identified by Daniel Goleman. This section also includes some questions which help to determine the features of the social and living aspects, as well as the psychological mood,

which affect the smoothness of the adaptation of repatriate students to the new conditions of life and the educational process as a whole, as well as success in acquiring English language skills in particular.

The third section concerned linguistic difficulties in all types of activities that students encountered in English training at the university. In this section authors developed a survey, which comprised both open and closed questions, to elicit the most exact answers for each question. Each question had answers that included the following options: 1. Completely agree, 2. Agree, 3. Don't agree, 4. Don't agree at all, and 5. Find it difficult to answer. The questions focused on the participants' own state identity, the degree to which the participants had integrated into the Kazakhstani community, and the respondents' opinions of the Kazakh, Russian, and English languages. This investigative approach helped to identify a number of problems encountered by oralman students learning English.

## 3. Results and Discussion

Data analysis in this study was ongoing throughout the data collection period. The ongoing analysis helped to identify the difficulties in teaching and learning of oralman students.

### 3.1. The Results of Quantitative Data Research

After data collection, the Cronbach's alpha estimate of internal consistency for the questionnaire was 0.870. Considering that the coefficient is more reliable when calculated on a scale above twenty items and is considered good at values between 0.80 and 0.90, we asserted the credibility of the survey (Table 4).

**Table 4.** Reliability statistics.

| Cronbach' Alpha | N of Items |
| --- | --- |
| 0.870 | 73 |

Certain data in our study processed with the program SPSS-22 in order to reveal valid correlations between explored variables. For correlation analysis the following variables were identified: anxiety (situational and individual); emotional self-regulation; emotional self-motivation; shyness; social support; resilience; cognitive flexibility (reassessing ability); English language test results. The correlation matrix for research variables was first constructed using SPSS software to examine the associations among all variables, as well as their relationship to components of the mentioned constructs with correlations being significant at the 0.01 level (2-tailed)—** and 0.05 level (2-tailed)—*.

Among the most significant correlations extracted by the system, we have outlined the following: 1. Positive correlation between resilience and cognitive flexibility (0.750 **), that may be interpreted as validation of strong connection between variables; 2. Positive correlation between situational anxiety and shyness (0.560 **), that speaks to their deep link; 3. Positive two-sided correlation between emotional self-motivation and availability of social support (0.490 **). Emotional responsiveness is a basic element of social support, while at the same time the last decreases psychological responses to stressors the oralmen suffer from and helps them to be resilient in problematic situations. Positive correlations were also excluded among cognitive flexibility, resilience, social support, and English language test results. The findings suggest that in the course of studying students may be generally getting more realistic in their expectations about social rules and norms of modern society, regardless of ethno-cultural characteristics. At the same time, the perceptions of the students in the educational context for the communication style should be positively encouraged. The research findings may indicate the difference in weight that each of the variables carries at the adjustment process in each case individually.

*3.2. The Results of Qualitative Data Research*

The results of the test taken by oralman students in order to identify their level of English revealed that 48% of respondents were at the elementary level, 33.6% were at the pre-intermediate level, and 18.4% of the respondents tested at the intermediate level, as shown in Table 5.

**Table 5.** English level test results of student repatriates.

| Levels | Intermediate % | Pre-Intermediate % | Elementary % |
|---|---|---|---|
| Speaking | 10.7 | 29.8 | 59.5 |
| Listening | 10.5 | 30.8 | 58.7 |
| Reading | 24.6 | 36.6 | 38.8 |
| Writing | 12.1 | 40.7 | 47.2 |
| Lexical and grammatical test | 34.4 | 29.8 | 35.8 |

As evidenced by the test outcome, university students in their second or third year generally test at the intermediate and pre-intermediate levels. The research group focused on first-year students. According to the test, 48% (288 students of 600) were at an elementary level, which confirms a low level of English proficiency that does not meet university requirements. The test results revealed that the absence of adequate audio and pronunciation samples in linguistic materials explained the students' poor speaking and listening skills.

The 64.9% of respondents (187 of 288) who tested at the elementary level had many grammatical inconsistencies in their speech and these students made mistakes when constructing sentences, phrases, and various language structures. In addition, these students exhibited a weak understanding of communicative intentions in counter-questions.

For 61.8% (178 of 288) of the respondents at the elementary level, one of the most difficult speech activities appeared to be listening. These students have poorly developed phonetic abilities; they could not identify rapid speech, and it was difficult for them to recognize the pronunciation of many unfamiliar words, which prevented adequate understanding of the material.

Among these students, 44.4% (128 of 288) did not reveal the skills of skimming and searching through a text, critical reading, and reading aimed at identifying details in written texts. Almost half of these students, 135 of 288 (46.8%), had difficulties in spelling words and correctly applying grammatical structures. The vast majority of respondents (80.9%, i.e., 233 of 288 respondents at the elementary level) did not know how to use different tense forms of verbs.

Cultural and social priorities and the focus on the Kazakh-Russian socio-cultural space largely determine language preference among oralman students. Different views on the concepts of nation-state, national characteristics, interests, traditions, language situations, and the educational process determine the types of obstacles that the students must overcome. The integration of ethnic Kazakhs into the new society of Kazakhstan inevitably leads to contact between two languages: Kazakh and Russian. The Russian language occupies a strong position in Kazakh society, not only because of the ethnic composition of the population, but also because of historical realities, particularly the influence of the long-standing language policy on the educational system. Consequently, the Russian language has a dominant position in all areas of communication in Kazakh society. Immigrant students' new living conditions actualize the issue of Russian language use and the degree of language mastery, which depends on a variety of extra-linguistic factors. In this regard, language is a particular difficulty for oralman students from China, Mongolia, Tajikistan, and Uzbekistan, the majority of whom are Kazakh speaking.

Data on the usage of the Kazakh and English languages, the specifics of their use in meaningful spheres of communication, the assessment of the importance of language ownership, and the need to study these languages indicates a strengthening of the position of the Kazakh language in the communicative language space of the country and an increase in the prestige of knowing English and the motivation to learn English.

According to the interviews and results of the survey, students are clearly aware of the importance of learning English. About 65% of the oralman students (390 students) are familiar with the State Programme of Development and Functioning of Languages in the Republic of Kazakhstan for the period 2011–2020 [30] and support the policy of trilingualism. Respondents believe that the trilingual policy has a positive effect on the harmonious development of personality (72.6%), the development of trilingualism will strengthen the position of the Kazakh language (68%), and the development of trilingualism strengthens the position of English (61%).

The responses reflect the opinions of the oralman students regarding the development of these languages: (a) the Kazakh language will inevitably prevail in all spheres of social life (89.1%) and (b) the dominance of the Kazakh language will improve the position of the English language (48%). The vast majority of respondents believe that the state's support of the English language is the correct policy. According to the results of the study, for 76.6% of the oralman students, English is an important course of study in addition to basic academic subjects. According to the respondents, (a) the English language is becoming increasingly popular (83.5%), (b) knowledge of English is prestigious and is a modern requirement for competitive specialists (67.3%), (c) knowledge of the English language is beneficial in economic terms (78.5%), (d) the English language provides opportunities to obtain good jobs (82.6%), (e) English can serve as a means of familiarizing oneself with the international community (84%), (f) the use of English in various fields has increased over the past five years (from 35.5% to 61.1%), (g) knowledge of English is a sign of the competitiveness of the country (76.1%), and (h) English should be studied from the first grade (79.8%). Of the respondents, 77% want to use English in their daily lives to watch television, news, and movies; participate in online correspondence; and read literature in the original language. Thus, the majority of respondents attribute the future of their professional and private activities to their knowledge of English. Proficiency in English is perceived as one of the primary factors related to career opportunities in Kazakhstan.

Overall, study results indicate that the full range of challenges that oralman students encounter in the course of learning English can be divided into three main groups. Oralman students must become accustomed to the new educational system. Of the respondents, 40% mentioned differences in educational and pedagogical cultures as one of the barriers to their successful mastery of English. They include mismatch of their language skills to the requirements at the university and a small number of hours of English per week. Due to the considerable workload, there is no time for oralman students to engage in further training, and they do not have enough resources to attend courses in English.

1. The system of teaching English in Kazakhstan differs from the systems of the countries from which these students immigrated. These differences can be found in educational materials, subjects, duration of lessons, and number of lessons per week. Consequently, the level of readiness (communication competence) of oralman students to understand the information they are taught is not in accordance with Kazakhstan high school requirements:

> *"I studied English as an elective course once a week. The lesson lasted 90 min with a 60-min break. We did not have Oxford or Macmillan books; the teacher used Chinese books".*

> (Gulzhamal, 22 years old, from China)

2. Listening and speaking difficulties: Many words are hard for them to pronounce and, as a result, are not remembered, some grammatical topics are not clear, as there are no similar structures in Kazakh.

> *"During the lesson when my teacher asks a question in English, I first think in Kazakh how to answer it and then start thinking of using the correct tense. It would be better if the teachers explained the grammar in Kazakh. Our teacher speaks and explains the grammar in English since our group's level is upper intermediate".*

> (Yelnur, 19 years old, from Mongolia)

*"I like to make up a dialogue in pairs. For example, Gulzhan helps me to ask a question, to answer it, and to use the right words. I feel her support when I am talking in a dialogue. However, when the teacher is talking to all students, I feel fear and try not to take part".*

(Ilan, 19 years old, from China)

The interview showed that students do not work on listening because they often do not follow the listening procedure. Many students (67 of 90 interviewees) answered that they listened to English songs in their free time but did not work to understand the song's lyrics.

Of the 90 interviewees, 73 indicated that they encountered difficulties in the reading process because of poor vocabulary, lack of mastery of reading techniques, and an inability to create sentences. Reading is extremely time-consuming for these students. The need to constantly look up words in dictionaries (Russian-English/Kazakh-English) discourages them from finishing a reading task.

As a consequence of these limitations, the oralman students rarely used English in everyday speech. Of the 90 interviewees, 28 answered the question, "How often do you write in English?" by saying that they sometimes used English to write messages and greeting cards.

Of the 90 interviewees, 53 noted that difficulties with pronunciation are primarily associated with a variety of specific differences in the articulation of sounds between the Kazakh and English languages. However, the oralman students' speech is also influenced by the phonetic features of the host country's language (Chinese, Mongolian, or Uzbek).

3. Difficulties related to pronunciation of sounds:

*"I feel shy to say the words in English. The English taught at my school differs from the English spoken by my groupmates. I do not want to be mocked".*

(Tansholpan, 19 years old, from China)

*"It is difficult for me to pronounce some words. I do not know how to pronounce some sounds, for example [ɵ]- [ð] width—with and [ǽ]- [ə] man – women. Listening is difficult for me".*

(Kuatbek, 23 years old, from Mongolia)

An important objective of this study was to learn about English teachers' views on how they can help oralman students become proficient in English. Eleni Oikonomidoy noted that immigrant students are struggling to learn a new language, in a new educational system and a new culture, while simultaneously filtering out the influences that interfere with being successful academically. Kazakh teachers also think that oralman students face the same problems and want to find out what difficulties they have in learning English [40].

The interviewed teachers emphasized the difficulty of teaching in mixed ability groups (local students and oralman students) because many of the oralman students have not learned English before, or they are at a lower level of English. Oralman students have problems with grammar, mispronounce some English sounds such as [p], [f], [ch], [sh], and it is difficult for teachers and local students to understand their accent. One of the teachers noted that in grammar, they make mistakes in gender, tense, and plurals. Oralman students make mistakes when using the personal pronouns *he* or *she*; they omit *will* in the future tense; they say *two girl* instead of *two girls.*

To summarize, oralman students and their teachers share the same opinions about the problems they have in learning and teaching English. Another problem includes the absence of English-Kazakh Dictionaries. It makes this work almost impossible. These problems reduce the motivation of students, so a way out of this situation needs to be found, both for the successful implementation of the language policy of the Republic, and for the improvement of the overall educational level of the youth.

## 4. Conclusions

Overall, the results of the study indicate that the full range of challenges that oralman students encounter in the course of learning English can be divided into three main blocks:

1.  Cultural difficulties are related to the mentality of different cultures (in our case, several cultures: English, Kazakh, Russian, Chinese, Mongolian, and Uzbek) and mastering the socio-linguistic and socio-cultural components of communicative competence. Features of the traditions and customs of the countries, from which oralman students came to Kazakhstan, are reflected in the formation of their views and attitudes to surrounding reality.

2.  Linguistic problems are divided into phonetics, grammar, and word formation spheres. There are structural features of articulation in the first language such as: an inability to distinguish the phonemes [i] and [r], a lack of the sound [h], reading sounds [d] and [t] aspirated, special pronunciation of the sound [r], differences in reading the sounds of the alphabet, and writing some of the sounds as combinations of letters. These inconsistencies with the Kazakh language bring some difficulties to teaching English. Grammatical difficulties are also based on the absence of similar structures in the Kazakh language: it has no category of gender and therefore there are no pronouns equivalent to *she* and *he*; the third person is the only person passed through the pronoun *ol*. The absence in the Kazakh language of verbs: ligaments, direct transfers to many prepositions, and articles complicates the process of learning. The word order in the sentence, the construction of interrogative sentences, and the use of the category of time affect the formation of communicative skills and complicate speech activity. Word formation in English is based on suffixes and prefixes. The Kazakh language has no prefixes, and this aspect creates a certain difficulty in the translation of words.

3.  Social-psychological: The immigration of ethnic Kazakhs to their historical homeland assumes that these oralmen can overcome the difficulties of the adaptation and integration processes. These difficulties include such social problems as desocialization (rejection of old norms and stereotypes, adoption of new forms of social and economic behavior), lack of objective information, problems with employment in one's field because of inconsistencies with the employer's requirements, leading to employment in low-skilled, low-paid work, and a lack of their own properties to live in. Faced with these issues, oralman students feel insecure because they are in an uncertain situation and consider themselves strangers. Oralman students must become accustomed to the new educational system. Of the respondents, 40% mentioned differences in educational and pedagogical cultures as one of the barriers to their successful mastery of English. The system of teaching English in Kazakhstan differs from the systems in the countries from which these students immigrated. These differences occur in the educational materials, subjects, duration of lessons and number of lessons per week. Consequently, oralman students' readiness to understand the information they are taught (communication competence) is not congruent with Kazakhstani high school requirements.

Overcoming language barriers and successful language adaptation are possible for oralman students arriving from different countries only through study of the cultural, linguistic, and behavioral norms and stereotypes [41]. One way of creating positive opportunities for them is to understand their life experiences and their realities. As the respondents noted, teachers play a crucial role in supporting the educational development of oralman students. Padilla A.M. (2006) noted that teachers not only have to be knowledgeable about their students' language, literacy, and academic needs, but also be well informed about second language learning and teaching [42,43]. In addition, without cultural understanding teachers will limit their capabilities in fostering growth and academic enhancement of oralman students [44]. According to Ali Borjian (2009), teachers who instruct non-English speaking immigrant students must be trained in language teaching pedagogy, including some knowledge of contrastive linguistics [45,46]. However, a comparable situation does not exist in Kazakhstan because English teachers in Kazakhstan are not proficient in oralmen languages. Thus, the methods of teaching English must be actively included in the processes of adaptation and integration of oralman students [47,48]. The professional experience of Kazakhstani universities' teachers promotes the development of various language skills in studying English. The research results indicate that

integrated methods for teaching English and the amplification of new and effective methods will enable oralman students to advance more rapidly in learning English.

## 5. Recommendations

The research indicated several factors English teachers should consider:

(a) The process of social adaptation is quite complicated for oralman students and there is a negative impact on their psychological state. Seeing and experiencing these problems at home, oralman students come to university with a negative emotional background, which affects the learning process. The socio-psychological adaptation of oralman students includes relief from stress and emotional pain in the course of learning English. The oralmen require gradual habituation to the English lesson requirements imposed by the university.

(b) The differences in cultural norms between Kazakhstan and countries of origin, from which oralman students come, affect the formation of new moral principles and cultural behavioral patterns. Linguistic and cultural learning aspects in the English classroom introduce the cultural peculiarities of the country of origin of the studied language, and an English teacher helps oralman students in the adaptation process through comparing them with the modern realities of Kazakhstan.

(c) Linguistic problems originate in the language barrier, which happens due to the oralman students' lack of knowledge in regard to writing in Kazakh. Oralman students who have arrived from Mongolia, China, or Afghanistan where they were using Arabic script, do not know the Cyrillic alphabet. English language teachers should pay attention to this aspect, considering the peculiarities of articulation, differences in the level of English, as well as the difference in the grammatical structure of the English language and the language of their country of origin.

(d) In order to resolve existing problems and improve the situation, the government implemented necessary projects and programs within the framework of the law on migration. Moreover, oralman students need moral support and understanding from the local population. In order to achieve positive dynamics in the teaching of English, teachers need to know and understand the difficulties faced by oralmen in the process of learning and everyday life.

(e) The creation of electronic dictionaries and grammar reference books from English to Kazakh should be considered.

(f) Putting an emphasis on the aspect of learning a language together with the culture, comparing cultural peculiarities of Kazakhstan and the countries of the target language when teaching English to oralman students is of importance.

(g) New types of programs for teaching groups of different levels need to be established.

Experience with teaching English to oralman students at the Eurasian National University suggests that the most efficient English teaching techniques are as follows:

(a) The use in the classroom of tasks created for this category of students to develop both receptive and productive speech activities. It would be efficient to create a multi-level system of exercises aimed at enhancing the use of aspects of language and the perception and generation of connected text;

(b) The use of the native language (in our case, the Kazakh language) in explaining English grammar. Such exercises enable students to understand the differences in the sociolinguistic principles of native and foreign languages and use them correctly in their speech;

(c) The introduction of such methods as blended learning, culture-based learning, and content-based learning in the process of teaching English;

(d) The use of a variety of teaching resources as additional sources that enhance the intercultural communicative competence of the students;

(e)   Providing opportunities for studying in separate groups to improve English skills, to create adaptation to a new culture, and to use all possible means of learning and teaching such as audio, video, and web resources.

## 6. Limitations of the Study

Limitations of this research should be considered. Firstly, according to the standard training program/curriculum, "A Foreign Language for Non-Linguistic Specialties of Higher Education Institutions for a Bachelor's Degree" (2010) [32], high school students need to be at the level of B1/B2. However, the experiment has exposed the oralman students' low levels of English. The structure of the four universities included in this research does not provide for split-level learning. Therefore, additional courses are not organized for elementary level students. Secondly, limitations of the study can be related to the limitations faced by the teachers in implementing the standard training program/curriculum, because cultural and social priorities and focus on the Kazakh-Russian socio-cultural space largely determine language limitations among oralman students.

The relevance of this study is determined by the need to intensify English teaching with the most efficient techniques and the education of oralman students in a social-pedagogical environment accounting for ethical standards. One example of these standards is that during the course of conducting research many oralman students were not given permission to be interviewed, as their parents lived in their countries of origin. Successful implementation of the standard training program/curriculum is dependent on developing a program for teaching groups of different levels and the moral support and understanding for oralman students' life experience and realities.

**Author Contributions:** For the present work, the contributions were distributed as follows: conceptualization, F.V.; data curation, G.T.; formal analysis, D.K.; investigation, J.S.; methodology, F.V. and J.S.; resources, G.T.; software, G.T.; writing—original draft, D.K.; writing—review & editing, F.V. and J.S.; supervision, J.S.

**Funding:** This research received no external funding.

**Conflicts of Interest:** No potential conflict of interest was reported by the authors.

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
