# Peer review of "The Socio-Linguistic Adaptation of Migrants: The Case of Oralman Students’ Studying in Kazakhstan"

_education, doi:10.3390/educsci9030164_

Round 1

Reviewer 1 Report

The article provides new knowledge as it tackles important issues related to the proccess of learning English by migrants in a multilingual context of the state actively promoting trilingual policy. Conclusions are sound and supported by data based on large number of questionnaires. The article should be of interest to various readers involved in the study of education, socio-linguistics and migration.

One area of discussions could be expanded. Oralman students repatriated from China, Mongolia,  Uzbekistan and other countries are influenced by the languages they have used as L1 or L2 in thos countries. Is there any specific interference in transition to Kazakh educational environment and English learning, depending on the language/culture of the country of origin? One comment to this matter can be found on page 10, with reference to phonetics. The general problem of cultural differences is mentioned in Conclusion 1, but the reader would appreciate more insight into the problem of the impact of both L1 and L2 on students' problems and acculturation in the English classroom in Kazakh education.

Also, the article requires a bit more attention to the quality of English. The text needs to be edited. Improvements ought to be made, e.g. “on the one hand” rather than “on the one side” should be used; or “not everyone adapt easily” should be corrected into “adapts”, the phrase “have misunderstanding in grammar” ought to be changed into “do not use grammar structures properly” or “do not use grammar structures properly” etc. There are other stylistic or grammatical flaws in English that require corrections, though in general the level of English is appropriate and the text is fully understandable.

The comment in lines 469-470 needs to be re-phrased because it fails to get the message across, also the very beginning of the Discussion section which follows. 

Author Response

Dear reviewer,

Thank you for perusal of our manuscript. We are immensely grateful for your benevolence and professional tact.

Response to Reviewer 1 Comments

Point 1.The article provides new knowledge as it tackles important issues related to the process of learning English by migrants in a multilingual context of the state actively promoting trilingual policy. Conclusions are sound and supported by data based on large number of questionnaires. The article should be of interest to various readers involved in the study of education, socio-linguistics and migration. 

One area of discussions could be expanded. Oralman students repatriated from China, Mongolia,  Uzbekistan and other countries are influenced by the languages they have used as L1 or L2 in those countries. Is there any specific interference in transition to Kazakh educational environment and English learning, depending on the language/culture of the country of origin? One comment to this matter can be found on page 10, with reference to phonetics. The general problem of cultural differences is mentioned in Conclusion 1, but the reader would appreciate more insight into the problem of the impact of both L1 and L2 on students' problems and acculturation in the English classroom in Kazakh education.

Response 1:  Thank you for your comment! The problem of linguistic and culture interference is one of the fundamental difficulties faced by the learners of second languages, in our case by oralman students. But we didn't aim to explore this aspect of adaptation. It is likely that at the next stage of our research we will touch upon these issues.

Point 2. Also, the article requires a bit more attention to the quality of English. The text needs to be edited. Improvements ought to be made, e.g. “on the one hand” rather than “on the one side” should be used; or “not everyone adapt easily” should be corrected into “adapts”, the phrase “have misunderstanding in grammar” ought to be changed into “do not use grammar structures properly” or “do not use grammar structures properly” etc. There are other stylistic or grammatical flaws in English that require corrections, though in general the level of English is appropriate and the text is fully understandable.

Response 2: The manuscript has been checked and edited.

Point 3.The comment in lines 469-470 needs to be re-phrased because it fails to get the message across, also the very beginning of the Discussion section which follows. 

Response 3: We rephrased the comment in the mentioned lines.

Thank you!

Reviewer 2 Report

p. 2, the last sentence of the first paragraph needs a reference. Also, it seems an 'and' was missing before low self-esteem.

p. 2 the first full paragraph is awkward. consider rewording to make sure it is syntactically accurate.

p. 2, citations are needed when the background and history of Oralman students were introduced.

p. 2, what reference was Table 1 based on? If it is based on the reference #8, then at the bottom of Table 1, a source needs to be listed.

p. 2, the last paragraph also needs literature support. All sounds the ideas/information from the author rather than from existing literature.

p. 2., second paragraph under literature review, line 111, the sentence is too long which makes the message being interrupted. Consider breaking down into separate sentences for clarity.

An independent paragraph needs to have at least two sentences.

p. 2, last paragraph/sentence before "English in Kazakhstan" needs citations. Again, one single sentence can't stand alone as a paragraph.

Section 1.2 English In Kazakhstan

The first two paragraphs need literature support. I understand that the author(s) are introducing the background, but such background should be supported by citations.

More importantly, the author(s) need to highlight the issue regarding English in Kazakhstan. In addition to the current situation, what is missing from the literature to inform practice? if it is connected with section 1.3, then I suggest combing the two and making it succinct.

p. 6 the last 3 paragraphs need literature support.

There is sufficient background of the study, however, what is missing is a strong rationale toward the end of the literature review. The author(s) have it; they just need to make it more cohesive and present the rationale in a clear manner to highlight the significance/contribution of this study.

2. Materials and Methods

Are these four universities similar to each other? Would there be any potential contextual factor that could impact the results? 

What were the research questions to be addressed in this study?

p. 7, regarding the survey, psychometric information seems to be missing. For example, what is the reliability of the placement test? Did the author use test items from an existing instrument? What were the items written based on in the third section? Did the author(s) develop the survey?

3. Results

p. 8, line 337, what were 'all subscales'? I might have missed it but the author(s) may want to specify all these subscales measured, making connection to my previous comment that information on how these subscales were developed is critical.

The results section is very lengthy and hard to follow. I suggest the author(s) to break it down by quantitative results (with statistical findings, preferably in a correlation table, in addition to descriptive tables of all the variables measured); followed by qualitative results where the interviews were to be shared. The section, in its current format, is not reader-friendly with too many texts without a break.  iThe author(s) may want to accommodate the needs of international readers of this journal.

4. Discussion

The discussion needs to be aligned with the literature reviewed earlier in this paper. There seems to be a disconnection. Consider combining discussion and conclusion.

Minor Observation

I suggest the paper to be edited by a native speaker to attend to grammatical and syntactic issues. 

This paper potentially has intellectual merit after a careful and thorough revision and rewriting are pursued. I hope my comments are helpful.

Author Response

Dear reviewer,

Thank you for perusal of our manuscript. We are grateful for your benevolence and valuable corrections. We have tried to heed all your comments and suggestions.  First, we expanded the list of references and added links. We also made some stylistic changes and other improvements.

Response to Reviewer 2 Comments

Point 1. p. 2, the last sentence of the first paragraph needs a reference. Also, it seems an 'and' was missing before low self-esteem; the first full paragraph is awkward, consider rewording to make sure it is syntactically accurate; citations are needed when the background and history of Oralman students were introduced.  

Response 1:  All necessary references were added to mentioned paragraphs; the end of the sentence has been corrected; the paragraph has been reworded and improved stylistically.

Point 2. p. 2, what reference was Table 1 based on? If it is based on the reference #8, then at the bottom of Table 1, a source needs to be listed; the last paragraph also needs literature support. All sounds the ideas/information from the author rather than from existing literature.  

Response 2:  All necessary references and a source were added. (http://rus.azattyq.org/content/kazakhstan-repatriants-statistics/)

Point 3. p. 2, second paragraph under literature review, line 111, the sentence is too long which makes the message being interrupted. Consider breaking down into separate sentences for clarity. An independent paragraph needs to have at least two sentences; p. 2, last paragraph/sentence before "English in Kazakhstan" needs citations. Again, one single sentence can't stand alone as a paragraph.

Response 3: We connected two paragraphs and divided the sentence to two separate sentences for clarity; the sentence has been taken out.

Point 4. Section 1.2 English In Kazakhstan

The first two paragraphs need literature support. I understand that the author(s) are introducing the background, but such background should be supported by citations; the author(s) need to highlight the issue regarding English in Kazakhstan. In addition to the current situation, what is missing from the literature to inform practice? if it is connected with section 1.3, then I suggest combing the two and making it succinct. 

Response 4: The references have been added; here we suppose that the information is represented clearly.

Point 5. There is sufficient background of the study, however, what is missing is a strong rationale toward the end of the literature review. The author(s) have it; they just need to make it more cohesive and present the rationale in a clear manner to highlight the significance/contribution of this study. 

Response 5: We have reformulated some of our statements in the literature review to highlight the contribution of this study.

Point 6. 2. Materials and Methods

Are these four universities similar to each other? Would there be any potential contextual factor that could impact the results? 

Response 6: The information about the differences has been added.

Point 7. What were the research questions to be addressed in this study?

Response 7:  Research questions have been formulated in a clearer form and added.

Point 8. p. 7, regarding the survey, psychometric information seems to be missing. For example, what is the reliability of the placement test? Did the author use test items from an existing instrument? What were the items written based on in the third section? Did the author(s) develop the survey? 

Response 8:  In our opinion, psychometric information is represented in the manuscript fully enough. The reliability is checked with the Cronbach's alpha. Nevertheless, clarification was added, that authors have developed a survey.

Point 9. 3. Results

p. 8, line 337, what were 'all subscales'? I might have missed it but the author(s) may want to specify all these subscales measured, making connection to my previous comment that information on how these subscales were developed is critical. 

Response 9:   We have clarified the information about the content of each section and added some additional information about psychological part.

Point 10. The results section is very lengthy and hard to follow. I suggest the author(s) to break it down by quantitative results (with statistical findings, preferably in a correlation table, in addition to descriptive tables of all the variables measured); followed by qualitative results where the interviews were to beshared. The section, in its current format, is not reader-friendly with too many texts without a break.  iThe author(s) may want to accommodate the needs of international readers of this journal.

Response 10:    The results section has been subdivided into two parts: with quantitative results and qualitative results.

Point 11. 4. Discussion

The discussion needs to be aligned with the literature reviewed earlier in this paper. There seems to be a disconnection. Consider combining discussion and conclusion.

Response 11. The Discussion has been disconnected into two parts. One part was added to the Results and the second one was included in Conclusion.  Literature review has the same points that were used in Discussion and therefore Literature review is remained without any changes).

Point 12. Minor Observation

I suggest the paper to be edited by a native speaker to attend to grammatical and syntactic issues. 

Response 12.  We hope now the manuscript “looks” better.

P.S. Since the authors of the manuscript live in different cities, there were some technical mismatches. Therefore, we highlighted the corrected or changed extracts in color.  

This paper potentially has intellectual merit after a careful and thorough revision and rewriting are pursued. I hope my comments are helpful.

Thank you so much!